# Implementation of the Publicly Funded Prenatal Screening Programme in Poland during the COVID-19 Pandemic: A Cross-Sectional Study

**DOI:** 10.3390/jcm11051317

**Published:** 2022-02-27

**Authors:** Bartosz Czuba, Jakub Mlodawski, Anna Kajdy, Dorota Sys, Wojciech Cnota, Marta Mlodawska, Sebastian Kwiatkowski, Pawel Guzik, Miroslaw Wielgos, Magda Rybak-Krzyszkowska, Anna Fuchs, Grzegorz Swiercz, Dariusz Borowski

**Affiliations:** 1Chair and Department of Gynaecology and Obstetrics, Faculty of Health Sciences in Katowice, Medical University of Silesia, 41-703 Ruda Śląska, Poland; bartosz.czuba@sonomed.net (B.C.); woytek@me.com (W.C.); 2Collegium Medicum, Jan Kochanowski University in Kielce, 25-369 Kielce, Poland; marta.mlodawska@ujk.edu.pl (M.M.); swierczag@poczta.onet.pl (G.S.); 3Department of Reproductive Health, Centre of Postgraduate Medical Education, 01-004 Warsaw, Poland; anna.kajdy@cmkp.edu.pl (A.K.); dorota.sys@cmkp.edu.pl (D.S.); 4Department Obstetrics and Gynecology, Pommeranian Medical University in Szczecin, 70-111 Szczecin, Poland; sebastian.kwiatkowski@pum.edu.pl; 5Clinical Department of Gynecology and Obstetrics, City Hospital, 35-241 Rzeszów, Poland; pawelguzik@gmail.com; 61st Department of Obstetrics and Genocology, Medical University of Warsaw, 02-015 Warsaw, Poland; miroslaw.wielgos@wum.edu.pl; 7Department of Obstetrics and Perinatology, University Hospital in Krakow, 31-501 Krakow, Poland; rybaczka@interia.pl; 8Chair and Department of Gynecology, Obstetrics and Oncological Gynecology, Medical University of Silesia in Katowice, 40-211 Katowice, Poland; afuchsuczelnia@gmail.com; 9Department of Perinatology, Gynaecology and Gynaecologic Oncology, The Faculty of Health Sciences, Collegium Medicum, Nicolaus Coeprnicus University, 85-067 Bydgoszcz, Poland; db1910@me.com

**Keywords:** prenatal screening, COVID-19, pregnancy

## Abstract

The COVID-19 pandemic in 2020 affected the entire healthcare system in Poland, causing medical personnel to be relocated to other duties and limiting patients’ contacts with healthcare professionals. A large part of the planned diagnostics and treatment was delayed due to lack of equipment and personnel. Against this background, we analysed the implementation of the publicly funded prenatal screening programme (PSP) in Poland compared to the previous year. This is a cross-sectional study. We used nationwide datasets on the implementation of the prenatal testing programme over the period 2019–2020, datasets from the Statistics Poland on birth and the data on the development of the COVID-19 epidemic in Poland. In the year 2020, we observed a 12.41% decrease in woman enrolled to the programme compared to 2019. However, the decrease concerned only women under 35 years of age. With respect to the number of deliveries in the calendar year, the number of patients enrolled in the programme decreased by 3% (31% vs. 34%, *p* < 0.001). We also observed an increase in estriol measurements per the number of patients included in the programme, and a reduction in the number of PAPP-A tests in the first trimester, which proves an increased share of the triple test in the prenatal diagnosis of chromosomal aberrations. With respect to the number of deliveries, the number of amniocentesis procedures performed under PSP decreased by 0.19% (1.8% vs. 1.99%, *p* < 0.0001). In 2020, compared to the previous year, the number of patients included in the prenatal testing programme in Poland decreased. In terms of the number of births in Poland, the number of integrated screening tests also decreased, at the expense of increasing the percentage of triple tests. There were also significant reductions in the number of invasive diagnostic tests.

## 1. Introduction

In 2020, the world was engulfed in the COVID-19 pandemic, which has since exerted a huge impact on all branches of medicine. The reasons behind it were the relocation of resources and healthcare workers (HCWs) within the public healthcare system, a change in the structure of the system by converting many departments to centres exclusively treating patients with SARS-CoV2 infections, the suspension of some scheduled treatments and an increase in HCW absenteeism due to disease and quarantine [1]. 

Countries around the world have initiated procedures intended to curb the spread of the virus and the effects of the disease caused by it. Due to the lack of treatment regimens and data on long-term complications, planned hospital admissions and outpatient visits have been limited. Pregnant patients who attend standard appointments with doctors and midwives during their pregnancy are in a particularly difficult position. In order to minimise the risk of patients becoming infected during transport or as a result of contact with health care professionals, the American College of Obstetricians and Gynaecologists (ACOG) recommended organizing some of the visits in the form of online doctor consultations [2].

The World Health Organization (WHO) recommends eight visits during the course of a pregnancy. The number of contacts with the doctor may vary depending on the condition of the mother and child. It has been proved that prenatal visits reduce the risk of pregnancy complications, intrauterine foetal death as well as infant death in the neonatal period [3]. According to the recommendations of the Polish Society of Gynaecologists and Obstetricians (PSGO), performing ultrasound examinations on schedule is the key aspect in the care of pregnant women. The first and second trimester prenatal tests, recommended for all pregnant women, are particularly important [4]. Under the public funding system (NFZ), these tests are reimbursed for a selected group of patients, in particular for women aged 35 or older and exposed to an increased risk of chromosome aberrations in the child [5]. 

The reimbursement covers the integrated screening test (IST) in accordance with Foetal Medicine Foundation (FMF) standards, including ultrasound examinations with an assessment of PAPP-A proteins in the serum and the free beta-HCG subunit, as well as risk calculation for non-hereditary chromosomal aberrations. In the case of patients who reported having an embryo with a crown–rump length (CRL) greater than 84 mm during pregnancy, the payer reimburses the serum triple screening test (STS), in which the concentrations of alpha-fetoprotein (AFP), bHCG and free estriol are assessed, and a risk calculation of chromosomal aberrations taken. The programme also includes a second trimester ultrasound scan, genetic counselling in the case of abnormal test results, and invasive diagnostics. The programme does not include non-invasive prenatal testing (NIPT). 

The recommendations of the Ultrasound Section of the PSGO, regarding ultrasound scans performed in pregnancy during the epidemic, did not limit the need for examinations in the first and second trimester [6]; they only emphasized the appropriate conditions for their performance to reduce the potential of SARS-CoV2 virus transmission between patients and medical personnel. However, surveys conducted among women indicate that, during COVID-19, almost half of them (47.41%) had part of their pregnancy visits conducted in the form of online doctor consultation [7].

We have not found any empirical data in the literature on how the COVID-19 pandemic influenced the implementation of prenatal testing programmes on a national scale; the published studies are concerned only with individual centres or networks of hospitals.

The aim of our study is to analyse the implementation of a publicly funded prenatal screening programme (PSP) in Poland during the SARS-CoV2 pandemic, and an assessment of how the pandemic year affected this implementation compared to the last year before the pandemic.

## 2. Material and Methods

### 2.1. Study Design

This was a cross-sectional study. The STROBE guidelines were employed in order to properly present the report from this study [8]. 

### 2.2. Setting and Data Sources

The study used nationwide datasets on the implementation of the prenatal testing programme over the period 2019–2020. The data were obtained from the National Health Fund at the request of the Ultrasound Section of the PSGO. In addition, the publicly available datasets from the Statistics Poland on birth rates [9] and the data on the development of the COVID-19 epidemic in Poland, available at https://ourworldindata.org/coronavirus (accessed on 12 October 2021), were used [10].

### 2.3. Participants and Study Size

The data pertaining to 237,396 participants of the prenatal screening programme during the target period were analysed. The participants were divided into two groups with regard to the date when the examination was performed: before the outbreak of the COVID-19 pandemic, i.e., in 2019, and during the COVID-19 pandemic, i.e., in 2020. 

### 2.4. Variables

The variables used in the analysis were all medical datasets available in the national register regarding the prenatal testing programme, including the number of procedures, tests, and consultations performed: ultrasound examinations of the first (11–13 weeks + 6 days of pregnancy) and the second trimester (18–22 weeks of pregnancy), PAPP-A quantification, estriol quantification, genetic counselling, amniocentesis, and trophoblast biopsy. The resultant risks of chromosomal aberrations were divided into three categories: low risk (<1:1000); medium risk (1:300—1:1000); high risk (>1:300). The analysis included medium- and high-risk categories. The main endpoint was the confirmation of a foetal birth defect based on ultrasound or invasive examination. Out of the datasets of the demographic yearbook, cumulative data on the number of births were used, divided by the age of patients, who were analysed in two categories: patients aged 35 and over and those under 35 years of age. 

### 2.5. Statistical Analysis

In the analysis, we took into account the absolute difference between the number of consultations provided, the ratio of the absolute number of consultations performed, and the number of patients enrolled in the programme, as well as the ratio of the absolute number of consultations performed to the number of births in a given year. We used the ‘N-1’ chi-squared test [11] to calculate the statistical significance of the difference in proportions. A difference was considered statistically significant if *p* < 0.05. We used the RStudio package (Version 1.2.1335) to calculate the results. 

## 3. Results

In 2020, 110,844 pregnant women were enrolled in the Prenatal Screening Programme (PSP). We observed a decrease of 12.41% women enrolled compared with 2019. This change was visible in the number of consultations, but the decrease in consultations concerned only women up to 35 years of age. In this group, a drop of 34.22% was observed. Among the patients aged 35 and over, an increase of 25% in the number of consultations was observed. The ratio of patients enrolled in the programme changed in favour of women over 35 years of age, who accounted for 36.8% of patients participating in the programme in 2019; while in 2020, they accounted for 52%, which is a statistically significant change (*p* < 0.001) (Table 1).

In 2020, 356,540 children were born in Poland (355,309 were live births and 1231 were stillbirths) and, in 2019, 376,192 children were born (374,954 live births and 1238 stillbirths), which represented a 5.3% decrease in deliveries in the year 2020 compared to 2019 [12]. With respect to the number of deliveries in the calendar year, the number of patients enrolled in the programme decreased by 3% (31% vs. 34%, *p* < 0.001). With respect to the number of deliveries in 2020, we observed an increase in the number of second-trimester ultrasound examinations by 0.96% (22.38% vs. 21.42%, *p* < 0.0001). The number of examinations in the first trimester decreased by 0.03%, but the difference in proportion was not statistically significant (*p* = 0.0537), while the number of biochemical tests performed in the first trimester of pregnancy decreased statistically significantly (Table 2). The number of PAPP-A determinations included in IST in relation to the number of ultrasound examinations in the first trimester decreased by 1.54% compared with 2019 (99.69% vs. 98.15%, *p* < 0.0001), while the ratio of estriol tests to the number of deliveries and to the number of patients enrolled in the programme increased. On the national scale, with respect to the number of deliveries, the number of amniocentesis procedures performed under PSP decreased by 0.19% (1.8% vs. 1.99%, *p* < 0.0001). In 2020, 1023 amniocentesis procedures fewer than in 2019 were performed under the programme nationwide.

We also observed changes in the ratio of the number of individual procedures to the number of patients enrolled in the programme. The percentage of patients in the programme who had their ultrasound examination performed in the first trimester increased by 6.47%, while the percentage of patients who underwent an ultrasound examination in the second trimester increased by 8.53%. The differences were statistically significant (*p* < 0.0001). The percentage of amniocentesis procedures and chorionic villus sampling in relation to the number of patients included in the programme did not change in a statistically significant way (Table 3). 

In Table 4, we present the results of PSP. Based on the first trimester examinations, the percentage of patients qualifying for the medium- and high-risk of chromosomal aberrations did not change (the difference for the high-risk group bordered on statistical significance). A greater percentage of children were confirmed with genetic abnormalities using invasive prenatal diagnosis (an increase of 0.07%, *p* = 0.00014). On the other hand, the relative number of foetal malformations in which a congenital defect was detected without invasive examinations, based on ultrasound examination alone, decreased by 0.8% (*p* = 0.0001). 

## 4. Discussion

The COVID-19 pandemic in Poland in 2020 occurred on a massive scale. During this year, 1.3 million cases (34,258 per million population (PMP)) of SARS-CoV2 infection and 28,554 deaths (755 PMP) were recorded. Overall, the hospitalization rate was 449 PMP, and the excess mortality rate was 1531 PMP. In Poland, however, we can observe a considerable underestimation of the disease due to the relatively small number of tests performed. Throughout the entire year 2020, this amounted to 183.6/1000 people, which is a figure almost 2.5 times lower than in neighbouring Germany (419/1000 inhabitants), and more than 3 times lower than in neighbouring Slovakia (594/1000 inhabitants) [10]. The first case of COVID-19 was officially found in Poland on 4 March 2020, and a state of pandemic was announced two weeks later. The beginnings of the pandemic, before the introduction of immunization, were special due to the high level of fear in society, related to interactions with healthcare workers [13]. A considerable number of scheduled diagnostics and non-emergency treatment cases were suspended [14]. Against this background, we analysed the operation of a publicly funded prenatal screening programme (PSP). From the payer’s perspective, the eligibility for the programme in 2020 did not change. However, we noticed a decrease in the number of patients participating in the programme per the number of deliveries in a given year. However, this decrease concerned only those patients aged under 35. It is difficult to determine the cause of such a change in proportions from a medical perspective. During the pandemic, the guidelines qualifying patients for screening tests in pregnancy did not change [15]. Already in April 2020, the guidelines of the International Foetal Medicine and Surgery Society emphasized the need for continued first trimester screening for chromosomal aberrations within the IST, as well as the continuation of screening for congenital anomalies in the second trimester, and invasive tests in the cases where indications occurred (amniocentesis and chorionic villus biopsy). The guidelines also emphasize the great benefits of some intrauterine procedures, such as intrauterine transfusions, thoraco-amniotic shunting, twin–twin transfusion syndrome (TTTS) laser therapy, and the surgeries combatting spina bifida [15,16], recommending their continuation, which additionally emphasised the necessity of early diagnosis.

The decrease in the number of female patients in the programme may be partially explained by the imposed isolation or quarantine during the period when the test can be performed. The time window is especially important for IST in the first trimester, which cannot be performed when the embryo is larger than 84 mm, i.e., during the first 13 weeks and 6 days of pregnancy [17]. In this case, in the era of high sensitivity testing, NIPT tests are recommended. For patients who decide not to have NIPT (mainly for financial reasons), a serum triple screening test (STS) is available under the programme. In the presented statistics, the triple test is reflected by the number of estriol concentration quantifications in the pregnant serum, as it is a quantification unique for STS. We observed an increase in the percentage of estriol quantifications under the programme in 2020 compared with 2019. We also observed a statistically significant reduction in the ratio of PAPP-A protein determinations (which is an integral part of IST) in relation to the number of ultrasound examinations performed in the first trimester, which may confirm the hypothesis that patients were referred or reported too late for the first trimester screening. 

The basis of STS is the use of only the pregnant woman’s serum. Ultrasound examination is not necessary for its performance. This results in less contact with HCWs and therefore less exposure to potential virus transmission. However, this test is not currently the gold diagnostic standard for chromosomal aberrations due to its lower sensitivity compared with the double test (77% vs. 93%) [18]. The use of STS without an ultrasound examination may entail an incorrect assessment of the gestational age during risk calculation. Moreover, this test does not allow for an early diagnosis of congenital anatomical defects. The patients who followed the recommendations and decided to have the NIPT performed are not included in PSP. 

In 2020, a significant decrease in amniocentesis procedures performed per number of deliveries was noted. This is a trend that we have observed worldwide. In a multicentre research trial carried out in Italy from March to May 2020, the average number of procedures performed in four centres decreased by over 20% in comparison with the same period in 2019 [19]. A similar analysis was conducted in a Turkish centre, in which, from11 March 2020 to 30 June 2020, the number of invasive diagnostic procedures decreased by one third when compared with the same period before the pandemic [20].

A clinical geneticist should be consulted each time after receiving any abnormal results of prenatal screening or before implementing invasive diagnostic testing procedures. This is one of the elements of medical counselling that can be carried out as part of telemedicine procedures [21,22]. Studies demonstrate that this model is effective and appreciated by patients during the pandemic. Approximately three quarters of patients were willing to switch from a face-to-face to an online consultation with a clinical geneticist due to the fear of contracting the virus [21]. In 2020 in Poland, we observed an over 3.5-fold increase in the percentage of genetic consultations per number of patients included in the programme. 

The fear of coming into contact with HCWs, as well as its stigmatization as a source of SARS-CoV2 infection, may be partly responsible for the decrease in the number of consultations provided in 2020. This hypothesis may be supported by the fact that there was a drastic decrease in the percentage of patients aged under 35 included in the programme. These patients may adhere to a false belief that the problem of non-congenital chromosomal abnormalities as well as birth defects affects only older women. Therefore, presumably, they can afford the comfort of not undergoing additional tests that might potentially increase exposure to SARS-CoV2 in the era of the pandemic. Meanwhile, fertility-wise, in most cases, it is the pregnancies of women under 35 years of age that are diagnosed with Down’s syndrome [23]. The pandemic is a period that has left a special psychological mark on pregnant women. Studies point to an increased risk of depression and anxiety disorder [24], in connection with the current epidemiological situation. Such disorders may be conducive to avoiding additional diagnostics, for which the contact with HCWs is necessary. 

Our analysis is limited by the lack of data concerning the quantity of tests and consultations provided by the private sector. In Poland, such elements of outpatient care are not recorded in the central database. During the peak waves of the pandemic, hospitals were overcrowded and HCWs was delegated to other activities, which could have affected the outflow of patients to private centres. It appears that most patients not included in the programme use this form of healthcare, and we can reason about it indirectly. In a 2019 questionnaire survey, 97% of female respondents demanded accessibility to prenatal testing irrespective of their social, ethnic, or religious background [25]. However, the available data do not allow the assessment of the testing accessible in the private sector. Taking into consideration the entire year 2020 creates yet another limitation. The period of the pandemic lasted for 9 out of 12 months in 2020; therefore, this provides us with an incomplete picture of the implemented changes; moreover, the partial data for particular months are not available. The growing popularity of NIPT tests may also affect the results. Poland is a country with a relatively small number of pregnant women who undergo NIPT [26], yet the popularity of this kind of testing grows year by year. NIPT is a highly sensitive test, but it does not work as a replacement for ultrasound or invasive diagnostic testing. The data concerning the impact of increasing NIPT on the number of amniocentesis procedures are contradictory. In some countries, after NIPT was introduced, the number of invasive tests decreased [27], whereas, in the remaining ones, it remained at the same level [26]. In our study, we observed a slight downward trend in the number of amniocenteses per number of deliveries, with a high probability that this change can be attributed to the increasing share of NIPT, but direct research on this matter is necessary.

The downward trend in the number of deliveries in Poland in each subsequent year constitutes a further limitation. In order to reduce the margin of error, the number of consultations was normalised by the number of deliveries in the calendar year. Nonetheless, if the duration of pregnancy is considered, this is merely an approximate coefficient. 

## 5. Conclusions

In 2020, the number of patients participating in the publicly funded prenatal screening programme decreased. The number of diagnostic procedures per number of patients included in the programme increased. When calculated per number of deliveries, a lower number of invasive procedures were performed under the programme. 

## Figures and Tables

**Table 1 jcm-11-01317-t001:** Data on the implementation of the prenatal testing programme in individual years.

Year	2019	2020	Difference (2019–2020)
Number of deliveries in Poland	376,192	356,540	−19,652
Number of women participating in the programme	126,552	110,844	−15,708
Number of women participating in the programme/number of deliveries [%]	33.64%	31.10%	−2.54%
<35 years of age	79,979	52,609	−27,370
≥35 years of age	46,573	58,235	11,662
Procedures performed under the programme			
Ultrasounds in total	167,230	163,098	−4132
First trimester ultrasound	86,931	83,307	−3624
Second trimester ultrasound	80,299	79,791	−508
PAPP-A quantification	85,324	83,050	−2274
Estriol quantification	1621	1639	18
Genetic counselling	43,909	41,598	−2311
Amniocentesis	7448	6425	−1023
Trophoblast biopsy	360	351	−9

**Table 2 jcm-11-01317-t002:** The number of procedures performed under the programme per number of births in a given year.

Year	2019	2020	Difference (2019–2020)	*p*
Procedures performed under the programme	n/number of deliveries in a given calendar year	n/number of deliveries in a given calendar year		
Ultrasounds in total	45%	46%	1.14%	*p* < 0.0001
First trimester ultrasounds	23.18%	23.37%	0.19%	*p* = 0.0544
Second trimester ultrasounds	21.42%	22.38%	0.96%	*p* < 0.0001
PAPP-A quantification	22.68%	23.29%	0.61%	*p* < 0.0001
Estriol quantification	0.43%	0.46%	0.03%	*p* = 0.0537
Genetic counselling	3.20%	11.67%	8.47%	*p* < 0.0001
Amniocentesis	1.99%	1.80%	−0.19%	*p* < 0.0001
Trophoblast biopsy	0.10%	0.10%	0.00%	*p* = 1

**Table 3 jcm-11-01317-t003:** Number of procedures performed under the programme per number of patients enrolled in a programme.

Year	2019	2020	Difference (2019–2020)	*p*
Procedures performed under the programme	n/number of patients in the programme	n/number of patients in the programme		
Ultrasounds in total	132%	147%	15.00%	
First trimester ultrasounds	68.69%	75.16%	6.47%	*p* < 0.0001
Second trimester ultrasounds	63.45%	71.98%	8.53%	*p* < 0.0001
PAPP-A quantification	67.42%	74.93%	7.51%	*p* < 0.0001
Estriol quantification	1.28%	1.48%	0.20%	*p* < 0.0001
Genetic counselling	34.70%	37.53%	2.83%	*p* < 0.0001
Amniocentesis	5.89%	5.80%	−0.09%	*p* = 0.3512
Trophoblast biopsy	0.03%	0.02%	−0.01%	*p* = 0.1266

**Table 4 jcm-11-01317-t004:** Prenatal screening programme results.

Year	2019	2020	Difference (2019–2020)	*p*
Risk 1: 300–1:1000/number of tests in the first trimester	5.91%	5.71%	0.20%	*p* = 0.0777
Risk >1:300/number of tests in the first trimester	7.53%	7.27%	0.26%	*p* = 0.0488
Confirmation of foetal anomalies in ultrasound examination (without invasive procedures)/number of patients in the programme	1.52%	0.71%	0.81%	*p* < 0.0001
Confirmation of foetal anomaly on the basis of invasive examination/number of patients in the programme	0.25%	0.33%	−0.07%	*p* = 0.0014

## Data Availability

The data that support the findings of this study are openly available in OSF Storage at https://doi.org/10.17605/OSF.IO/ECYU9 (accessed on 10 December 2021).

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
