# Peer review of "Implementation of the Publicly Funded Prenatal Screening Programme in Poland during the COVID-19 Pandemic: A Cross-Sectional Study"

_jcm, 2022, doi:10.3390/jcm11051317_

Round 1
Reviewer 1 Report
The topic addressed is current but not new. Despite the short time that Covid-19 was discovered, much research has been done on the subject, also in terms of sexual and reproductive health. The decrease in pregnancies, births, prenatal care etc. has already been addressed in different studies in several countries. The article should address some important issues.
The study is not justified. The objective is stated in a confusing way "to analyze the implantation" with the current results only described but not analyzed.
There is no authorization report from any ethics committee. The description of the characteristics of the sample is limited only to age, which dichotomizes it into less than 35 years and over 35 years. Age is a continuous variable. The results are not presented clearly, nor are the tables clear or aesthetic.
Among the strengths of the research is its large sample size. Limitations are reported but no reference is made to methodological limitations, for example the quality of the data, since it is a national database there is a lot of variability in the completion of the records, this has not been addressed. Therefore, a misclassification bias may exist. Also, as the authors say, the private system is not controlled. Due to the pandemic, public centers were saturated and this could have caused many women to use private services given the difficulty of accessing public services, the influence of this situation on the results has not been addressed either.
The conclusions are logical, but they do not respond to the objective since they do not analyze, although they coincide with that of the multiple studies published on the subject, there was a reduction in pregnancies, childbirth and prenatal care.
The bibliography is disordered, it does not follow any style. Not referenced homogeneously
Author Response
Thanks with the quick reply. Below, we answer all the point-by-point remarks. All changes to the manuscript are marked with the tracking of changes option.
“The topic addressed is current but not new. Despite the short time that Covid-19 was discovered, much research has been done on the subject, also in terms of sexual and reproductive health. The decrease in pregnancies, births, prenatal care etc. has already been addressed in different studies in several countries. The article should address some important issues.”
Thank you for this remark, in our opinion this paper is address some important issues. Indeed, the impact of the COVID-19 pandemic on prenatal care has already been assessed, but no study has ever done this at a national level and this is the main strength of our study. All previous studies assessed prenatal care at a local level in one facility or in a network of centers. In our opinion, the work has great value as it evaluates the prenatal screening program at the national level what can implication for extrapolation for other nations.
“The study is not justified. The objective is stated in a confusing way "to analyze the implantation" with the current results only described but not analyzed.”
In our opinion study is justified. According to Oxford Dictionary definition of the word “analyze” is: to examine the nature or structure of something, especially by separating it into its parts, in order to understand or explain it. In our study, we described the data and then analyzed it by categorizing it, normalizing it in relation to the number of births in a given year, comparing subsequent years by performing statistical tests assessing the statistical significance of the difference, and then critically analyzing the obtained results in a discussion. So we cannot agree with the statement that "results only described but not analyzed"
“There is no authorization report from any ethics committee.”
According to polish law a cross-sectional study based on anonymous data that is publicly available, i.e. data obtained from the insurer and data from reports published by the national statistical institute, does not require the consent of the bioethics committee, because there is no intervention in the study, the data is an anonymous and retrospective.
“The description of the characteristics of the sample is limited only to age, which dichotomizes it into less than 35 years and over 35 years. Age is a continuous variable. The results are not presented clearly, nor are the tables clear or aesthetic.”
Age is a continuous variable but has been categorized according to the usual standards. In prenatal diagnostics, we usually operate with a dichotomous variable resulting from tradition and the fact of the exponential increase in the risk of chromosomal aberrations from the moment the patient is 35 years old. [Morris JK, Mutton DE, Alberman E. Revised estimates of the maternal age specific live birth prevalence of Down's syndrome. J Med Screen 2002; 9:2.]
“Among the strengths of the research is its large sample size. Limitations are reported but no reference is made to methodological limitations, for example the quality of the data, since it is a national database there is a lot of variability in the completion of the records, this has not been addressed. Therefore, a misclassification bias may exist. Also, as the authors say, the private system is not controlled. Due to the pandemic, public centers were saturated and this could have caused many women to use private services given the difficulty of accessing public services, the influence of this situation on the results has not been addressed either.”
We consider this remark very valuable. We have modified the "discussion" section in the article in accordance with the reviewer's recommendations. We have added information about the limitations resulting from the issues raised by the reviewer. PWe are indeed unable to assess data quality completely, but this is a problem that affects all multicentre studies, the so-called “sampling error”. However, the property of sampling error is that increasing the sample size tends to reduce the sampling error. With our sample size, this error is marginal. We are confident about the completeness of the data contained therein, because the completeness of the research report is necessary for the payment from the program, which strongly acts as a motivating factor for reporting for centers. The aim of the work from the beginning was to evaluate the public sector, information on the performance of research in the private sector has been provided only to gain a broader perspective for reader.
The conclusions are logical, but they do not respond to the objective since they do not analyze, although they coincide with that of the multiple studies published on the subject, there was a reduction in pregnancies, childbirth and prenatal care.
We have already argued our opinion on the analysis. We normalized our data by dividing the quantitative data by the number of deliveries, such normalization eliminates the influence of the total number of deliveries in a given year on our analysis.
The bibliography is disordered, it does not follow any style. Not referenced homogeneously
A very valid point. We have corrected the bibliography to follow the style of the journal.
Reviewer 2 Report
ReviewerIn the manuscript titled “Challenges in the implementation of the publicly funded prenatal screening programme in Poland during the COVID-19 pandemic: a cross-sectional study”, the authors compared the implementation of the publicly funded prenatal screening program in 2020 in Poland to the previous year (2019). The topic could be of interest; however, several concerns need to be raised:
1- There is a lack in highlighting gaps in current understanding or conflicts in current knowledge about the subject of the manuscript in the introduction. Also, the need for investigations in the topic area is not demonstrated. Please add it.
2- Please add a conclusion to the abstract.
3- Page 2, line 70, WHO should be written entirely.
4- The hypothesis of the study has been missed.
5- It would be informative to add the exact dates of the study (the starting and ending months of including data)
6- Were there any inclusion or exclusion criteria for the data collection?
7- Page 3, line 120, I suggest writing “(11-13 + 6 weeks of pregnancy)” as (6 and 11-13 weeks of pregnancy).
8- Page 3, line 151, What did the authors exactly mean by “year over year”? Please clarify If they meant from 2019 to 2020.
9- The legend in table 1 is not appropriate.
10- Table 2, the second row needs to be modified.
11- What do the authors mean by “programme results” in legend table 4. Do they mean “prenatal screening program results”? Please make it clear.
12- Page 6, line 205, COVID-19 has been a pandemic, not an epidemic.
13- Please add suitable references to Lines 195-200.
14- The authors need to clarify “TTTS” in line 223.
15- The discussion is too long and lacks depth (it is purely descriptive). Please remove unnecessary redundancy.
16- The title of the study is not matched with the manuscript. The authors analyzed the data more than dealing with challenges
Author Response
Thank you very much for the revision below, we will answer the points raised point by point.
“ReviewerIn the manuscript titled “Challenges in the implementation of the publicly funded prenatal screening programme in Poland during the COVID-19 pandemic: a cross-sectional study”, the authors compared the implementation of the publicly funded prenatal screening program in 2020 in Poland to the previous year (2019). The topic could be of interest; however, several concerns need to be raised:
1- There is a lack in highlighting gaps in current understanding or conflicts in current knowledge about the subject of the manuscript in the introduction. Also, the need for investigations in the topic area is not demonstrated. Please add it.”
Very valuable information, we have added the necessary paragraph in the text.
“2- Please add a conclusion to the abstract.”
Corrected
“3- Page 2, line 70, WHO should be written entirely.”
Corrected
“4- The hypothesis of the study has been missed.”
We added the necessary information in the last paragraph of the "introduction" section
“5- It would be informative to add the exact dates of the study (the starting and ending months of including data)”
The study covered the entire year 2019 and 2020, so we did not provide specific dates.
“6- Were there any inclusion or exclusion criteria for the data collection?”
The data used was used in its entirety, and the inclusion and exclusion criteria are the same as the exclusion and inclusion criteria for the prenatal testing program, which are presented in the "introduction" section.
“7- Page 3, line 120, I suggest writing “(11-13 + 6 weeks of pregnancy)” as (6 and 11-13 weeks of pregnancy).”
We corrected as You suggested.
“8- Page 3, line 151, What did the authors exactly mean by “year over year”? Please clarify If they meant from 2019 to 2020.s”
We corrected as You suggested.
“9- The legend in table 1 is not appropriate.”
We have changed the legends to table 1.
“10- Table 2, the second row needs to be modified.”
We have corrected this passage as recommended.
“11- What do the authors mean by “programme results” in legend table 4. Do they mean “prenatal screening program results”? Please make it clear.”
We have corrected this passage as recommended.
“12- Page 6, line 205, COVID-19 has been a pandemic, not an epidemic.”
We have corrected this passage as recommended.
“13- Please add suitable references to Lines 195-200.”
We have corrected this passage as recommended.
“14- The authors need to clarify “TTTS” in line 223.”
We have corrected this passage as recommended.
“15- The discussion is too long and lacks depth (it is purely descriptive). Please remove unnecessary redundancy.”
We shortened the discussion as recommended.
“16- The title of the study is not matched with the manuscript. The authors analyzed the data more than dealing with challenges”
We changed the title as suggested.
Reviewer 3 Report
"Challenges in the implementation of the publicly funded prenatal
screening programme in Poland during the COVID-19 pandemic: a cross-sectional
study." - is an interesting paper in which the aim is to analyse the impact of the COVID-19 pandemic on prenatal diagnosis in pregnant women.
The analysis is presented in an interesting and thorough manner. Very interesting is the analysis of the frequency of consultations carried out in the age groups of pregnant women below and above 35 years. The authors have shown that during the 2020 pandemic, patients aged over 35 years attend diagnostic consultations more frequently than before the pandemic, in contrast to patients aged under 35 years.
However, there was no analysis of whether these were first or subsequent pregnancies for these patients and, if they were subsequent pregnancies, how the previous pregnancies were managed.
It would also be worth extending the analysis of this trend to subsequent years in the future.
We also cannot exclude the possibility that patients over 35 years of age, being more mature and experienced, were better able to predict the potential risk of abnormal pregnancy development.
It would also be very valuable if the paper could show how the number of prenatal interventions carried out on the fetus for diseases such as heart defects, spina bifida, diaphragmatic hernia or TTTS syndrome changed during the first year of the pandemic. Conducting such an analysis would greatly increase the value of the paper.
To my knowledge, prenatal interventions on the foetus have fallen by around 30% in 2020.
Author Response
Thank you for your quick review of our work. Below we will answer it point by point. We rolled wrote all the changes to the manuscript with the 'change tracking' option.
"Challenges in the implementation of the publicly funded prenatal
screening programme in Poland during the COVID-19 pandemic: a cross-sectional
study." - is an interesting paper in which the aim is to analyse the impact of the COVID-19 pandemic on prenatal diagnosis in pregnant women.
The analysis is presented in an interesting and thorough manner. Very interesting is the analysis of the frequency of consultations carried out in the age groups of pregnant women below and above 35 years. The authors have shown that during the 2020 pandemic, patients aged over 35 years attend diagnostic consultations more frequently than before the pandemic, in contrast to patients aged under 35 years.
However, there was no analysis of whether these were first or subsequent pregnancies for these patients and, if they were subsequent pregnancies, how the previous pregnancies were managed.”
Thank you very much for this remark, unfortunately the data on whether the patient is nullipara or pluripara is not available in the insurer's register. Therefore, such data is unavailable to us.
“It would also be worth extending the analysis of this trend to subsequent years in the future.
We also cannot exclude the possibility that patients over 35 years of age, being more mature and experienced, were better able to predict the potential risk of abnormal pregnancy development. “
We plan to continue the trend analysis in the coming years.
“It would also be very valuable if the paper could show how the number of prenatal interventions carried out on the fetus for diseases such as heart defects, spina bifida, diaphragmatic hernia or TTTS syndrome changed during the first year of the pandemic. Conducting such an analysis would greatly increase the value of the paper. To my knowledge, prenatal interventions on the foetus have fallen by around 30% in 2020.”
Unfortunately, in Poland intrauterine interventions are not financed from the prenatal screening program, but from a separate fund of the insurer, in addition, some of these interventions are not fully reimbursed in Poland, and are financed by funds from foundations and non-governmental organizations, therefore the data is not fully available for use.
Reviewer 4 Report
see attached file

Round 2
Reviewer 2 Report
The authors have satisfactorily responded to all questions and comments.
Just a minor comment: I think it does not need to use a separate paragraph for the conclusion in the abstract.
Reviewer 4 Report
I am not sure there is the need to specify the subparagraph "conclusion" into the abstract, given that other sub-sections have not been included. maybe the other reviewer intended that the the abstract was missing conclusive details. however, please look at and adhere to the journal guidelines in this regard.
In addition, look at eventual typos still present into the text during the proof corrections. I found two, but look after other eventual mistakes (page 6, discussion, line 209-210: after "two weeks later" there are two full stops (..) - page 8, line 287 after "indirectly" there is an extra-space) remembering that a well-written article is easily cited.